# Experimental Investigation of Pre-Flawed Rocks under Dynamic Loading: Insights from Fracturing Characteristics and Energy Evolution

**DOI:** 10.3390/ma15248920

**Published:** 2022-12-13

**Authors:** Guifeng Zhao, Lei Zhang, Bing Dai, Yong Liu, Zhijun Zhang, Xinyao Luo

**Affiliations:** 1School of Resource Environment and Safety Engineering, University of South China, Hengyang 421001, China; 2School of Physics and Optoelectronic Engineering, Shenzhen University, Shenzhen 518060, China; 3School of Resources and Safety Engineering, Central South University, Changsha 410000, China

**Keywords:** SHPB, flawed rocks, cracking mechanism, energy dissipation, rock-burst

## Abstract

Different fractures exist widely in rock mass and play a significant role in their deformation and strength properties. Crack rocks are often subjected to dynamic disturbances, which exist in many fields of geotechnical engineering practices. In this study, dynamic compression tests were carried out on rock specimens with parallel cracks using a split hopkinson pressure bar apparatus. Tests determined the effects of strain rate and crack intensity on dynamic responses, including progressive failure behavior, rock fragmentation characteristics, and energy dissipation. Based on the crack classification method, tensile–shear mixed cracking dominates the failure of rock specimens under the action of impact loading. Increasing the flaw inclination angle from 0°–90° changes the predominant cracking mechanism from tensile cracking to mixed tensile–shear cracking. The larger the loading rate, the more obvious the cracking mechanism, which indicates that the loading rate can promote the cracking failure of rock specimens. The fragmentation analysis shows that rock samples are significantly broken at higher loading rates, and higher loading rates lead to smaller average fragment sizes; therefore, the larger the fractal dimension is, the more uniform the broken fragments of smaller sizes are. Energy utilization efficiency decreases while energy dissipation density increases with increasing strain rate. For a given loading rate, the energy absorption density and energy utilization efficiency first decrease and then increase with increasing flaw inclination, while the rockburst tendency of rock decreases initially and then increases. We also find that the elastic–plastic strain energy density increases linearly with the total input energy density, confirming that the linear energy property of granite has not been altered by the loading rate. According to this inherent property, the peak elastic strain energy of the crack specimen can be calculated accurately. On this basis, the rockburst proneness of granite can be determined quantitatively using the residual elastic energy index, and the result is consistent with the intensity of actual rockburst for the specimens.

## 1. Introduction

An increasing demand of energy and resources means rock engineering plays an increasingly important role in the construction of deep underground spaces in which the defective rock is disturbed by the dynamic loads of blasting and impact points [1,2]. Static compressive loads are derived from gravitational stress and tectonic stress [3,4,5], and dynamic loads can be inferred from measurements, such as drilling, quarrying, and earthquakes, as shown in Figure 1. The most typical rock engineering involves the excavation of underground caverns in which the dynamic loads from blasting act on the rock column. Natural rock inherently contains many pre-existing natural defects which significantly affect the stability of underground engineering structures, so accurately obtaining the fracture behavior of stable rock containing pre-existing defects under the impact load is of vital importance for construction safety and rock engineering disaster prevention [6,7].

In the past few decades, people have devoted themselves to the study of dynamic strength and fracture behavior under shock loading. The dynamic strength of rock materials has been shown to have a pronounced rate-dependent resistance. So far, studies on defective stable rock have mainly focused on purely static loading conditions. A number of static uniaxial compression tests were performed on rocks containing single or multiple defects. It is very difficult to use classical fracture mechanics to study the failure mechanism of rocks due to the heterogeneity of rock materials. Therefore, laboratory tests, as a basic method, have been widely used to study the mechanics and fracture of rocks under various loading conditions [8,9,10,11].

For instance, a group at MIT led by Einstein systematically investigated the strength and fracturing of rock and rock-like (i.e., gypsum) materials containing pre-existing flaws under static compressions [12,13,14] and found great impacts on the strength, deformation, crack propagation, and coalescence of the rock under static loading. Wu et al. [15] studied the crack strength of sandstone material with two parallel inclined cracks under uniaxial compression and concluded that it mainly contains three modes of shear, mixed shear tension, and wing crack tension. Wong and Chau et al. [16] studied the effect of pre-existing elliptical defects on the deformation and fracture behavior of natural rocks under static uniaxial compression and concluded that the geometry of elliptical defects determines the principal stress and ultimate failure mode of the rock specimen and elliptical form defects weaken its compressive strength and elastic modulus. Dai et al. [17,18,19] studied the mechanical and energy dissipation characteristics of granite under cyclic impact loading and the results showed different axial pressures. The damage value criterion established based on energy dissipation could characterize the relationship between damage and the number of impacts, which showed a slow increase, a steady increase, and a high-speed increase, and the damage value depended mainly on the last impact. Zhang et al. [20,21] studied crack development and damage patterns under combined dynamic–static loading of parallel double fractured rocks and concluded that under a constant strain rate in the triaxial compression tests, the confining pressure effect is very obvious for the dynamic triaxial compressive strength and secant modulus and the 45° fractures represent high synergy between the slip rate of the cracks in the bridge zone and the dissipation energy generated by the specimens. Bobet et al. [22] and Li et al. [23] studied the mechanical behavior and dynamic cracking of single and multi-defect rocks under static uniaxial compressive loading tests and concluded that they are both significantly affected by the shape stability of geometric defects, including defect length, angle, spacing between the two defects, and the length and angle of the rock bridge. These results show that the mechanical behaviors of the defective rock under static and dynamic loading are different, but they are all significantly affected by the geometry of the defect, including its length, spacing, and angle. For static experiments on specimens with defects in 3D conditions, Li et al. [24] tested rock-like specimens with two and three parallel pre-existing fissures under uniaxial loading. In their study, seven types of crack coalescence were identified from the experimental results of specimens containing various fissure angles. However, although rock-like materials can simulate some behaviors of real rock, it is very difficult to simulate all properties of real rock materials, such as heterogeneity, mineral grains, boundary effect, and cementation. Li et al. [25] investigated the real-time cracking behavior of sandstone specimens containing two coplanar flaws with different flaw angles under uniaxial compression. The relationship between the coplanar flaw angle and crack coalescence stress was constructed in accordance with the photographic monitoring results. Furthermore, Yang et al. [26] tested the strength and failure behavior of sandstone specimens with a pair of unparallel flaws under uniaxial compression. They adopted photographic monitoring and the AE technique and analyzed crack coalescence and AE evolution behavior depending on various flaw angles.

In addition to static load conditions, dynamic loads, such as earthquakes, explosions, and rockbursts are also widely present in mining and geotechnical engineering. Under dynamic loading, it is recognized that the strength of the rock (i.e., compressive, tensile, and shear strength) depends on the loading rate; that is, rock strength increases with the loading rate [27,28]. Under dynamic loading, the propagation and crack growth rates of stress waves are much higher than those of static loading, resulting in different mechanical responses of rock specimens containing defects. Based on split Hopkinson bar SHPB equipment, Yan et al. [29,30] carried out a series of dynamic load tests on rock samples with a single defect and used a high-speed camera to analyze the fracture process of the samples. They found that the mechanical parameters of the defective specimens show obvious rate dependence under dynamic disturbance and the observed shear cracks are the main crack types under dynamic loading in an “X”-shaped shear failure. Zhang et al. [31] systematically studied the effects of strain rate and crack density on the mechanical properties and energy characteristics of rock specimens with parallel fractures and concluded that more mixed cracks would appear on the surface of the specimen with an increase in crack strength and that crack networks have become more complex. Both dynamic strength and deformation modulus showed a decreasing trend with the increasing defect strength. Yan et al. [32] and Fan et al. [33] studied the modeling problem of dynamic fractures in rocks and concluded that the Numerical Population Method (NMM) can be used to study the propagation of stress waves in fractured rocks. To investigate the effect of dynamic loading on crack propagation, Wu. et al. [34] used Split Hopkinson Pressure Bars (SHPB) to test rhombohedral marbles containing single defects with different dip angles, resulting in the existence of artificial defects that change the failure mode of marble under dynamic loading from the cracking of intact specimens to the shearing of defective specimens. Li et al. [35] studied the effect of a single pre-existing cavity (a circular or elliptical cavity) on the fracture behavior of marble specimens in SHPB experiments and concluded that shear cracks were more efficient than tensile cracks under dynamic loading conditions, which appeared earlier and dominated the fracture process. Zhou et al. [36] conducted dynamic tests on the behavior of cracking at the edge and proved that crack velocity is not constant and its propagation toughness is related to crack propagation velocity. In addition, Zhang and zhao [37] used a digital imaging technique (DIC) to measure the full-field strain on the surface of rock specimens under dynamic loading and showed the microscopic mechanism of brittle failure before the appearance of a macroscopic crack.

Previous studies have shown that subterranean rock masses with prefabricated fractures are often subjected to static compression and dynamic loads. However, although single defect or multiple defects are widely distributed in natural rock mass, no scholars have ever studied the mechanical behavior and energy consumption characteristics of parallel crack rocks with different fracture angles under different loading rates. Therefore, in this study, granite with prefabricated parallel double defects was introduced in the dynamic load test through an improved SHPB device to simulate the engineering rock with prefabricated defects under the action of a certain loading rate. The SHPB system is the first to study the effects of loading rate and fracture angle on the dynamic strength, deformation characteristics, failure mode, rockburst, and energy dissipation of fractured rocks. The fracture distribution of rock samples with parallel fractures under the loading rate and the failure modes of rocks at different fracture inclination angles are comprehensively analyzed through high-speed camera images. Section 2 first introduces the sample preparation and test setup and then evaluates the energy calculation method for the SHPB test system. Section 3 systematically expounds the experimental results, including dynamic stress balance, dynamic strength, rock progressive cracking behavior, rockburst law, and energy evolution. Section 4 comprehensively discusses some issues of crack classification and potential guidance for practical engineering. Section 5 discusses and interprets the discussion. Section 6 summarizes the entire study.

## 2. Experimental Setup

### 2.1. Specimen Preparation

We used granite for the experimental research due to its mechanical isotropy and structural homogeneity, and we made rock samples with parallel double fractures with different fracture angles. In order to obtain a clearer observation of crack evolution, a single granite with good geometric integrity and uniformity was cut to obtain prismatic samples with a height, width, and thickness of 45, 45, and 20 mm, respectively, which proved the feasibility and effectiveness of the prismatic rock samples in the SHPB test. Then, the high pressure water jet cutting machine (Changsheng stone firm, Ningbo, China) was used to cut the rock specimens with a length, width, and thickness of 10 mm, 1 mm, and 1 mm parallel cracks at different angles (0°, 45°, and 90°). The upper and lower surfaces of the sample were polished with a grinding machine (Haideli Machinery Company, Yueyang, China) during the processing of the sample preparation, and the processing accuracy strictly conformed to the ISRM (International Rock Mechanics) standard. The flatness of the surface was controlled at 0.05 mm, and the vertical deviation of the upper and lower surfaces was controlled at 0.25°. The samples were divided into complete samples and samples with parallel double cracks with different inclination angles. The average density of the samples was 2580 kg/m^3^, and the elastic modulus was about 9.86 GPa. A total of 60 samples were produced, as shown in Figure 2.

### 2.2. Experimental Apparatus and Techniques

SHPB devices in Central South University (Difeisi Dynamic High Voltage Technology Company, Nanjing, China) have been widely used to study the dynamic mechanical properties of rocks under different loading rates according to the recommendation of the International Society of Rock Mechanics. In this study, a SHPB device with a diameter of 50 mm was used, which consisted of an impact rod, an incident rod, and a transmission rod with lengths of 300, 3000, and 2000 mm, respectively. The stress transfer components included impact rods, incident rods, and transmission rods with lengths of 300, 3000, and 2000 mm, respectively. The entire system was composed of switches and safety valves and could generate a slowly rising half-sine wave to eliminate the wave oscillation effect and achieve a constant strain rate on brittle materials. All rods were made of ultra-high-strength titanium alloy with a density of 7800 kg/m^3^ and an elastic modulus of 211 GPa. A thin copper sheet connected to the free end of the incident rod acted as a pulse shaper, and two strain gauges were installed separately. The two strain gauges were used to record the strain signal during dynamic loading at the center of the incident rod and the transmission rod.

During the experiment, the sample was sandwiched between the incident rod and the transmission rod, and the interface between the rod and the sample was evenly coated with lubricant to reduce the friction effect. Once ready, the conical impact bullet was fired from the air gun at high velocity and struck the front end of the injection rod. The resulting incident wave propagated along the strip, and a part was reflected onto the incident rod when the wave reached the incident rod-specimen interface (reflected wave), while the rest (transmitted wave) passed through the specimen and was further transmitted to the transmitted rod superior. The magnitude of the transmitted wave depended on the difference in wave impedance between the rod and the specimen. The magnitudes of the incident, reflected, and transmitted waves were collected using strain gauges attached to the surfaces of the incident and the transmitted rods, and the collected wave forms were displayed on an oscilloscope. The strain rate, peak stress, and stress–strain curve of the specimen was obtained after processing the data based on the three-wave method.

The high-speed (HS) photography system was composed of an HS camera (Ketianjian Photoelectric Technology Company, Hunan, China), a pair of high-intensity flashes (Jiangsu Xuanlang Lighting Company, Yangzhou, China), and a high-performance laptop (Dell, TX, USA). The HS camera recorded images with a resolution of 128 × 256 fps pixels and an area of 50 × 100 mm^2^ at an inter-frame time of 6.5 μs. The pulse signal released by the oscilloscope triggered the HS camera when the strain gauge signal on the incident rod arrived so that the HS image could be synchronized with the dynamic loading stress. The HS camera was located at a safe distance of 1.5 m from the specimen to prevent damage caused by breaking rock fragments punching out, as shown in Figure 3.

### 2.3. Data Processing

One-dimensional stress propagation is still guaranteed for dynamic SHPB compression tests. According to previous studies, the baselines for incident, reflected, and transmitted waves are all zero. A strain gauge can record the entire strain signal, including the strain. The dynamic load at the end of the sample was determined using the Formulas (1)–(3). Based on the one-dimensional stress wave theory, mechanical parameters, such as the average dynamic stress, strain, and strain rate of the sample, can be calculated using the following formulas [38]:(1)$$\sigma_{\left(t\right)}=\frac{A_{e}}{2A_{s}}\left[\sigma_{I}\left(t\right)-\sigma_{R}\left(t\right)-\sigma_{T}\left(t\right)\right]$$
(2)$$\varepsilon \left(t\right)=\frac{1}{\rho_{e}C_{e}L_{s}}\int_{0}^{t}\left[\sigma_{I}\left(t\right)+\sigma_{R}\left(t\right)-\sigma_{T}\left(t\right)\right]d_{t}$$
(3)$$\overset{˙}{\varepsilon \left(t\right)}=\frac{1}{\rho_{e}C_{e}L_{s}}\left[\sigma_{I}\left(t\right)+\sigma_{R}\left(t\right)-\sigma_{T}\left(t\right)\right]$$
where $\sigma_{I}\left(t\right)$, $\sigma_{R}\left(t\right)$, $\sigma_{T}\left(t\right)$ are the incident stress, reflected stress, and transmission stress of the rod, respectively; $A_{e},\rho_{e},$ and $C_{e}$ are the cross-sectional area, density, and longitudinal wave velocity of the rod, respectively; and $A_{s}$ and $L_{s}$ are the cross-sectional area and length of the sample, respectively. In the SHPB impact test, the incident wave energy caused by the impact load is mainly converted into the energy of the reflected wave and transmitted wave as well as the energy absorbed by the rock sample. Incident energy (*E_I_*), emitted energy (*E_R_*), and transmitted energy (*E_T_*) can be indirectly calculated from the corresponding three stress wave signals using Equations (4)–(6) [39]:(4)$$E_{I}=\frac{A_{e}}{C_{e}\rho_{s}}\int_{0}^{t}\sigma_{I}^{2}\left(t\right)d_{t}$$
(5)$$E_{R}=\frac{A_{e}}{C_{e}\rho_{s}}\int_{0}^{t}\sigma_{R}^{2}\left(t\right)d_{t}$$
(6)$$E_{T}=\frac{A_{e}}{C_{e}\rho_{s}}\int_{0}^{t}\sigma_{T}^{2}\left(t\right)d_{t}$$

The energy absorbed by the rock specimen (*E_A_*), the energy utilization efficiency *N_d_* defined in this study, and the energy absorption density of the rock specimen can be determined as follows using Equations (7)–(9):(7)$$E_{A}=E_{I}-E_{R}-E_{T}$$
(8)$$N_{d}=\frac{E_{A}}{E_{I}}$$
(9)$$e_{d}=\frac{E_{A}}{V_{0}}$$
where *V_0_* is the volume of the rock specimen and the absorbed energy can be divided into three main groups: crack propagation and fracture and damage energy of micro-cracks in the sample, the kinetic energy of flying fragments, and other forms of consumption which can be neglected energy, such as heat and sound. The energy absorption rate can reflect the amount of energy absorbed by the rock sample during the dynamic loading process, while the energy absorption density represents the energy absorption of flawed rocks per unit volume.

### 2.4. Sieving Tests

Debris distribution is a key indicator in rockburst and rock excavation. In this study, rock fragments were collected in a plastic collection box after the shock loading experiment was completed. Analyses were performed using standard sieves with mesh sizes of 0.075, 0.25, 0.5, 1, 2, 5, 10, 20, and 40 mm,. After sieving, the rock was divided into fragments of different sizes. The cumulative mass distribution can be obtained by weighting each group of rock fragments. To quantify the rock fragmentation at different loading rates, the average size of the rock fragments can be determined using Equation (10):(10)$$d_{m}=\left(\sum w_{i}\bar{d_{i}}\right)/\sum w_{i}$$
where *d_i_* is the mean size of the fragments situated between two levels of mesh size and *w_i_* is the interval mass percent of the rock fragments corresponding to $\bar{d_{i}}$.

It is generally believed that rock fragments are similar, and fractal geometry theory is widely used in rock fragmentation analysis. In this study, the fractal dimension *D* is used to quantify the fragment size distribution, as shown in Figure 4, using Equation (11):(11)$$y_{i}=\frac{M(<d_{i})}{M}={\left(\frac{d_{i}}{d_{max}}\right)}^{3-D}$$
where *M (<d_i_)* is the cumulative mass of fragment size less than *d_i_*; *M* is the total mass of all fragments; And *d_max_* represents the maximum size of the rock fragments.

For the sieving test, the values of *d_i_* and *d_max_* were determined using the mesh of the sieve device and *M* and *M (<d_i_)* were calculated after sieving, so the fractal dimension *D* of the rock can be obtained using Equation (11).

The best fit can be derived using least squares linear regression through the data points in Figure 4d based on using fractal theory to quantify the blockiness of the rock after fragmentation and taking the logarithm of both sides of Equation (11). The slope of the linear fit line is 3–*D*, and its fractal dimension *D* can be further obtained.

## 3. Experimental Results

### 3.1. Dynamic Stress Equilibrium Check

Dynamic stress balance before failure is a prerequisite for a valid dynamic impact specimen and can be checked by comparing the dynamic stress across the specimen during the entire test period. Figure 5 shows the stress balance at both ends of the three test specimens under different loading rate conditions in the dynamic test, and the loading rate is represented by the slope of the straight line segment before the peak of the stress time history curve. The dynamic stress is the sum of the incident stress and the reflected stress at the incident end of the specimen, expressed as incident stress+ reflected stress. The dynamic stress is caused by the transmitted wave at the transmission end of the specimen, marked as transmitted stress. It is obvious that the dynamic stress on both sides is almost the same throughout the loading process which indicates that the loading experiment has achieved the stress balance on the specimen and the axial inertia effect can be neglected since there is no force difference in the specimen that causes inertial force.

### 3.2. Dynamic Deformation Characteristics

Figure 6 shows the dynamic stress–strain curves of rock specimens with different fracture angles. The results show that the strain rate significantly affects the shape of the dynamic stress–strain curve. The rock sample has no strain recovery under the action of each loading rate, and it increases monotonically during the entire loading process. The stress–strain curve of the specimen does not appear in the compaction stage under a certain loading, indicating that the granite has good compactness. The stress–strain curve includes three stages, including the elastic stage, the yield stage, and the post-peak failure stage. The the elastic stage and yield stage of the specimens develop faster as the loading rate continues to increase, but the yield stage shortens, mainly because the increasing loading rate leads to increasing linear elastic deformation which shortens the development process of linear elasticity and the yield stage in the curve.

## 4. Progressive Cracking Behaviors

### 4.1. Dynamic Fracture Process of Cracked Rock in Impact Test

The failure mode of the specimen is a key indicator to reveal the failure mechanism of the rock, and Figure 7 shows the different dynamic fracture processes captured by the high-speed camera. In this section, three typical fractured rock samples with different inclination angles were selected to study progressive cracking behavior under impact load. Only the dynamic fracture process of representative samples was selected due to a space limitation. For each specimen, five snapshots were selected covering the initial stage, the fracture initiation stage, the stable crack propagation stage, the peak stress stage, and the post-peak failure stage. Three typical crack types were highlighted in each diagram to illustrate the fracture mechanism of some local cracks more clearly. For each photograph, the left and right sides of the tested specimen were the transmitted bar and the incident bar, respectively.

Cracks are mainly divided into three types: tensile cracks, shear cracks, and mixed tensile–shear cracks. Tensile cracks are consistent with the loading direction of the specimen, while shear cracks expand in the oblique direction, and tensile–shear mixed cracks are common cracks in rock samples with parallel loading directions and vertical loading directions. The failure of the specimen is mainly caused by the initiation, expansion, and penetration of the crack at the tip of the parallel crack to both ends of the specimen, and it is connected to the tensile shear crack formed at both ends of the specimen. New cracks are formed under the action of a higher loading rate. It is generally accepted that the wing crack (primary crack) initiates at an angle with the pre-existing fissure and extends along the loading direction during the loading process of the pre-cracked specimens. However, secondary crack initiate after wing crack in two initiation directions, including coplanar with the fissure (coplanar secondary crack) or at an angle (oblique secondary crack), and one is similar to the wing crack but in the opposite direction, as shown in Figure 7.

Figure 8a shows the typical cracking behavior of the 0° crack under the loading rate of 2721.75 GPa/s. The first picture corresponds with the situation where the rock begins to load (0 μs), and the sample is not deformed at this time. Two tensile cracks (FT crack) begin to form in the middle and lower part of the parallel cracks when the rock is loaded for 99 μs, and no obvious cracks form at the crack tip. As the loading continues, mixed tensile shear cracks (MTS crack) occurr in the upper left and lower left corners of the specimen at 112 μs, and the two cracks expand diagonally and gradually connect with the middle tensile crack. When the rock loads to the peak stress (137 μs), two tensile–shear mixed cracks form in the upper and lower right corner of the sample, respectively, and the propagation speed is faster than the previous two. At the beginning, the tensile crack in the middle of the parallel crack continues to expand and connect with the crack at the left and right ends of the specimen. The MTS crack and the FT crack are connected and penetrated as the stress gradually decreases in the post-peak stage, and new tensile shear cracks (FS crack) are generated at the left and right ends of the sample. Finally, the specimen forms an “X”-shaped shear tensile failure.

Figure 8b shows the typical cracking behavior of the 45° cracked rock under the loading rate of 2397.57 GPa/s. The first picture corresponds to a situation where the rock starts to load (0 μs), and the sample is not deformed at this time. At 84 μs, a short tensile crack is formed in the central rock bridge region of the parallel crack, and a new crack begins to sprout at the tip of the crack. A shear crack (FS crack) occurs at the upper left corner of the sample as the loading continues, while mixed tensile cracks (MTS crack) appear at the upper and lower right ends of the specimen, and a shear back-wing crack appears at the tip of the crack, which gradually connects with the lower mixed tensile crack. The shear cracks at the upper and left ends of the sample extend to the parallel cracks along the diagonal of the specimen when the rock is loaded to peak stress (123 μs). At this time, a new shear crack appears at the upper right part and gradually expands along the crack tip. As the stress gradually decreases in the post-peak stage, a horizontally developed tensile crack appears in the upper part of the sample and connects with the shear cracks at both ends, resulting in the final tensile shear failure.

Figure 8c shows the typical cracking behavior of the 90° crack specimen under the loading rate of 2485.89 GPa/s. The first picture corresponds to the situation where the rock starts to load (0 μs), and the sample is not deformed at this time. The tensile cracks (FT crack) parallel to the loading direction are generated in the tip and middle of the crack when the rock is loaded at 92 μs. As the loading continues, new shear cracks (FS crack) occur at the upper and right ends of the specimen at 107 μs, and they connect wit the tensile crack generated at the tip of the crack, and the shear crack at the upper end continues to expand along the loading direction. The cracks around the crack tip connect to each other to form an “X”-shaped crack when the sample is loaded to the peak stress (130 μs), and the shear crack in the lower part of the sample gradually expands. New tensile cracks occur at the lower end of the crack and the lower right corner of the specimen as the stress gradually decreases to 53.9% of the peak stress, and this eventually results in an “X” shear tensile failure.

### 4.2. Final Failure Modes of Rocks with Different Fractures under Impact Loading

The final failure mode of the specimen under impact loading has important scientific significance and engineering application value for studying the crack propagation characteristics and the overall failure mechanism of rock mass. Three main failure modes are classified according to the final dynamic fracture modes and the positional relationship between the new and pre-cracked cracks, as depicted in Table 1.

Faliur type I is axial splitting tensile failure along the loading direction of the specimen, mainly for 0° parallel crack specimens. The macroscopic failure crack of the sample generally expands along the loading direction parallel to the direction of the crack inclination angle due to the increase in the impact load, and a composite tensile shear crack occurs at the crack tip. The initial macroscopic cracks in the main diagonal directions (two and four quadrants) and the far-field cracks at the top and bottom of the specimen surface usually propagate through the surface along the loading direction to the complete failure of the specimen.

Faliur type II is “X” type” shear failure, mainly for specimens with 45° parallel cracks and crack specimens under the 2357.97 GPa/s loading rate. The failure modes are rock bridge penetration and crack initiation at the crack tip X-shaped shear failure extending in the diagonal direction. The macroscopic tensile crack is generated at the top and bottom of the crack, and the macroscopic initial crack formed at the loading section generally extends along the parallel end face. The secondary and far-field cracks in the main diagonal direction propagate through the entire surface of the specimen along the loading direction until it fails.

Faliur type III is tensile–shear composite failure mode, mainly samples with 90° parallel cracks. Shear cracks first occur at the tip of the crack of the sample, and with the increasing loading rate, the shear crack gradually develops from a local shear crack distributed at and near the tip of the crack to a tensile extension extending from the tip of the crack along the top and bottom of the sample. The crack then propagates along the loading direction and penetrates toward the incident and transmission end, respectively, resulting in the tensile–shear composite failure of the specimen.

In addition to newly created cracks, some small rock debris may be exfoliated from the behavior of the failed specimen. Figure 9 shows the final failure modes of all fractured specimens, where the solid symbols, half-filled symbols, and hollow symbols indicate Type I (axial splitting tensile failure), Type II (“X” type shear failure), and Type III (tensile–shear hybrid failure) rock failure mode. In our tests, the Type I failure mode is the most common, and the percentage of each failure mode is about 38%, 33%, and 13%. The Type I failure mode becomes more obvious as the loading rate increases, which means the loading rate plays an important role in the failure mode, and each crack angle contains three failure modes, which indicates that the crack angle has no obvious effect on its failure mode.

The cumulative mass classification curve of the sample can be obtained via the sieving test. The results show that the rock loading rate has a great influence on the failure mode of the sample. Regardless of the size of the crack inclination, the rock fracture increases with the increase in the loading rate, but the specimens remained intact or slightly split at low loading rates. However, the rock is broken into large strips or fragments under the action of a medium loading rate. The rock sample is crushed into small fragments or even powder under the action of a high loading rate. The cumulative mass percentage gradually increases to 100% as the fragment size increases from 0.075 mm to 40 mm, as shown Figure 10.

Table 2 and Figure 11 show the effects of loading rates and fracture angles on average rock fragment sizes and fractal dimensions. The fractal dimension is smaller at a lower loading rate, while the factual dimension increases as the loading rate increases to a higher level, and its increasing speed gradually slows down, and its maximum value can reach 2.1. For a given loading rate, the fractal dimension decreases with an increasing fracture angle and reaches a maximum when the fracture angle is 0°. The average fragment size for most specimens remains in the range of 19–24 mm. The average fragment size generally decreases as the loading rate increases, and the higher the mass fraction of small-sized fragments, the faster the decline rate; therefore, it can be inferred that a higher loading rate can cause a more significant effect within the rock specimen and produce a smaller average size. In general, both the loading rate and the fracture angle have certain effects on the rock crushing characteristics.

## 5. Energy Dissipation

### 5.1. Energy Utilization Efficiency and Energy Absorption Density

Figure 12 shows the effect of loading rates and fracture angles on the energy utilization efficiency and energy absorption density of the rock samples, and the table summarizes the energy distribution of the fractured samples. Under the same fracture inclination angle, the energy utilization efficiency of the fractured rock samples is between 20% and 40%, and the energy absorption rate decreases with the increasing loading rate, regardless of the size of the fracture angle, as shown in Table 3 and Figure 10. It can be speculated that most of the energy will not be absorbed by the rock when the incident energy is not enough to break the rock and that the energy utilization efficiency is very low. The applied dynamic load is not so large for rock breaking engineering. Instead, there is a reasonable threshold, and using a certain method to control the loading level effectively improves the energy utilization rate. On the contrary, the figure shows that the loading rate promotes the energy absorption of the rock sample, and the absorbed energy per unit volume increases gradually. Specifically, the energy absorption density increases from 0.6 J/m^3^ to 1.4 J/m^3^ as the loading rate increases from 2200 GPa/s to 3100 GPa/s. In addition, for a given loading rate, the absorbed energy per unit volume and energy absorption rate first decreases and then increases with the increasing crack angle. For example, when the loading rate is 2400 GPa/s, the energy absorption density corresponding to each crack specimen is 0.92, 0.63, and 0.88 J/cm^3^, respectively. This decrease–increase trend is somewhat like that of dynamic strength due to the fact that higher strength means more energy will be dissipated for rock failure.

### 5.2. Rockburst Proneness of Pre-Crack Specimens

#### 5.2.1. Rockburst Characteristics of Pre-Crack Specimens at Different Loading Rates

Rockburst is an extremely violent and destructive phenomenon, usually accompanied by a large number of rock fragments ejected in a short period of time; therefore, high-speed camera systems are used to record the “rockburst phenomenon”. The rockburst phenomena of fractured rocks with different angles at different loading rates are similar. First, the rock particles are ejected or chipped, and then the cracks gradually expand with the increasing loading road, and the fragments on the rock are separated from the sample, and finally, a large amount of rock fragments and powders are produced, and a loud cracking sound is emitted. There is a strong rock fragment ejection phenomenon during the failure process for all fractured rocks with small loading rates. The rockburst tendency is lower for samples with higher loading rates, while the ejection velocity of rock blocks is lower, and the spray distance is short. It is worth noting that the duration of rock macroscopic failure increases significantly at higher loading rates. This indicates that the loading rate promotes the initiation of micro-cracks in the rock, thereby delaying the occurrence of rockburst. However, the micro-defects of the rock sample increase significantly due to the higher loading rate, which reduces its deformation ability and prolongs the stable failure stage of the sample.

In addition, quantitative analysis of rockburst phenomenon of fractured rock under impact load is still the most critical issue. In deep rock engineering, comprehensive judgment is often made based on the distance, distribution range, and acoustic properties of rock fragments ejected during rockburst. The ejection mass ratio outside the lever, *M_E_* (mass ratio of rock fragment outside the lever to total exfoliated rock fragment, defined by Gong et al., 2018b), was used as a quantitative index to evaluate the degree of rockburst occurrence. *M_E_* reflects the distance and mass of the ejected rock fragments and also reflects their kinetic energy (to a certain degree). The classification of *M_E_* is [40]
(12)$$M_{E}=0,\;\left(No\;rockburst\;proneness\right)$$
(13)$$<M_{E}\leq 0.4,\;\left(Low\;rockburst\;proneness\right)$$
(14)$$0.4<M_{E}\leq 0.6,\;\left(\mathrm{Medium}\;\mathrm{rockburst}\;\mathrm{proneness}\right)$$
(15)$$M_{E}>0.6,\;\left(\mathrm{High}\;\mathrm{rockburst}\;\mathrm{proneness}\right)$$

The statistical results of *M_E_* for the specimens with cracks are presented in Figure 13. The *M_E_* tends to increase as the loading rate increases, and the high rockburst area is concentrated in the loading rate of 2600 GPa/s–3100 GPa/s. The results show that the loading rate has a significant effect on the rockburst under the impact load. The inter-particle distances within the rock decrease as the loading rate increases and the interaction between particles enhances, which creates additional stress. The *M_E_* of the rock sample is lower than 0.6 when the loading rate is lower than 2600 GPa/s, while the occurrence of rockburst is not obvious, and medium rockburst occurs. The reason is that the mineral composition and structure inside the rock are gradually destroyed under a certain impact load, resulting in a significant reduction in the releasable elastic energy of the rock and weakening of the rockburst strength. The degree of rockburst decreases first and then increases as the fissure inclination angle increases, and the magnitude relationship is 90° > 0° > 45°; however, the *M_E_* of 90° fractured rock suddenly increases under the action of a large impact. It can be seen that the fracture angle is not obvious for judging the rockburst tendency, but *M*_E_ could accurately reflect the rockburst tendency of rock under different loading rates.

#### 5.2.2. Quantitative Evaluation of Rockburst Proneness of the Pre-Crack Specimens under Loading Impact

It is well-known that the occurrence of rockburst is an outward expression of residual elastic energy release inside rocks. Higher rock fragment ejection velocity and rockburst propensity corresponds to larger elastic residual elastic energy. Therefore, it is extremely important to calculate the residual energy accurately during the impact process of cracked specimens with respect to energy storage and dissipation.

It is assumed that the granite sample follows the first law of thermodynamics during the impact process; in other words, the sample does not interchange energy with the outside, and the energy of the rock only comes from the impact of the compression level on the specimen. Energy density is used as a characterization of strain energy results to eliminate the effect of the sample volume. Therefore, the strain energy variations of the granite specimen can be described as:(16)$$u_{t}=u_{d}+u_{k}$$
where *u_t_* is the total input strain energy density; *u_k_* is the releasable strain energy stored in the rock, i.e., elastic strain energy density; and *u_d_* is the energy used for compaction of pores, crack propagation, and plastic deformation inside the rock, i.e., the dissipated strain energy density.

Figure 14 shows the *u_k_* and *u_d_* of the fractured granite specimen under six loading levels are accurately calculated during the impact loading process, and the calculation formula of *u_t_*, *u_k_*, and *u_d_* are as follows:(17)$$u_{t}=\int_{0}^{\varepsilon_{1}}f\left(\varepsilon \right)d\varepsilon$$
(18)$$u_{e}=\int_{\varepsilon_{0}}^{\varepsilon_{1}}f_{u}\left(\varepsilon \right)d\varepsilon$$
(19)$$u_{d}=u_{t}-u_{e}=\int_{0}^{\varepsilon_{1}}f\left(\varepsilon \right)d\varepsilon -\int_{\varepsilon_{0}}^{\varepsilon_{1}}f_{u}\left(\varepsilon \right)d\varepsilon$$
(20)$$u_{a}=\int_{0}^{\varepsilon_{u}}f\left(\varepsilon \right)d\varepsilon$$
(21)$$u_{t}^{p}=u_{t}/v_{0},\;u_{d}^{p}=u_{d}/v_{0},u_{k}^{p}=u_{k}/v_{0},u_{a}^{p}=u_{a}/v_{0}$$
where $f\left(\varepsilon \right)$ and $f_{u}\left(\varepsilon \right)$ represent the loading curves; $\varepsilon_{0}$ represents the elastic strain point; and $\varepsilon_{1}$ represents the plastic deformation of specimen after loading.

According to Equations (17)–(21), we could obtain the $u_{d}^{p}$, $u_{k}^{p}$, and $u_{a}^{p}$ of fractured rock. The highly linear relations between $u_{k}^{p}$ and $u_{t}^{p}$, $u_{d}^{p}$, and $u_{t}^{p}$ are found during the shock loading process of fractured rocks by analyzing the results of these strain energies, as illustrated in Figure 15. These results show that the linear energy storage and dissipation laws can be used for dynamic loading experiments of fractured rock samples. Theoretically, external loads are a prerequisite for rock energy accumulation, and both elastic strain energy density and plastic strain energy density increase with increasing input energy density. Additionally, the elastic strain difficulty density first increases and then decreases with the increasing crack angle, while the energy density first decreases and then increases. The coefficient of determination of linear fitting for $u_{d}^{p}$ and $u_{t}^{p}$ are 0.99, 0.93, and 0.90, respectively, and the coefficient of linear fitting for $u_{k}^{p}$ and $u_{t}^{p}$ are 0.90, 0.98, and 0.97.

Based on these findings, we can accurately calculate the relationship between the strain energy densities at the peak strength (i.e., peak elastic strain energy density $u_{d}^{p}$ and peak plastic strain energy density $u_{k}^{p}$) of the specimens and the loading rate by using the linear storage and dissipation laws. Both the loading rate and the crack angle have a certain influence on the change law of the strain energy accompanying the impact. As shown in Figure 16, the input strain energy density first increases and then decreases with the increasing loading rate for the rock specimens with a 0° and 45° flaw, while the rock specimens with a 90° flaw are on the rise. The elastic strain energy density and plastic strain energy density of rock specimens with 0°, 45°, and 90° flaws increase continuously with the increasing of the loading rate during the whole loading process, and the elastic strain energy density is always greater than the plastic strain energy density, which indicates that the energy absorbed and stored inside the rock is mainly converted into the energy required for rock failure and fracture expansion.

$K_{s}=u_{d}/u_{k}$ is defined based on the elastic strain energy storage law, and *K_s_* is used to represent the stable state of the sandstone system under storage and dissipation. Figure 17 shows the changes in *K_s_* with the loading rate, where *K_s_* first increases and then decreases with the increasing loading rate, and the value of *K_s_* is always less than 1, which indicates that most of the strain energy of the rock is transformed into elasticity under impact loading. The specimen generally absorbs the energy during the impact loading, which also verifies the conclusion that the rock sample absorbs energy when rock-burst occurs. With the increasing of *K_s_*, the initiation and propagation of cracks are accompanied by the increasing of elastic strain energy. At this time, less plastic strain energy is generated and gradually released outward which reduces the ability of the rock to resist external loads. The system of the crack sample gradually changes from an unsteady state to a steady state, and the energy storage capacity also gradually weakens. The *K_s_* of the 45° and 90° fracture specimens begin to decrease with the increasing loading rate when the loading rate increases to 2500 GPa/s, and a large amount of strain energy is used for the sliding friction of rock fracture, which indicates that the closure of pores and micro-cracks is gradually completed. However, the elastic strain energy continues to increase, the entire rock system still maintains a stable state, the reversible elastic strain energy in the rock system gradually accumulates, the energy behavior continues to be energy storage, and its ability to store energy is further enhanced. As *K_s_* gradually decreases, the degree of steady state increases gradually while the stability increases gradually. *K_s_* reaches the minimum value under the action of large impact loading, and *u_d_/u_k_* is also the minimum value at this time, while the energy storage capacity of the sample gradually increases. The turning point occurs when the loading rate is 2800 GPa/s for the specimens with a 0° flaw, which indicates that the 0° crack rock has stronger impact resistance and needs more external energy to break the rock with the increasing loading rate, but its *K_s_* has the same variation law of specimens with the 45° and 90° rocks.

We can calculate the absolute residual energy of the rock at failure and instability (i.e., the residual elastic energy density, *A_EF_*, or the difference between $u_{d}^{p}$ and post-peak failure energy density $u_{a}^{p}$) when the rock is at failure and instability in order to verify the accuracy of the proposed rockburst classification method, as shown in Figure 17. We obtain the *A_EF_* (a new rockburst classification method) to evaluate the rockburst proneness of fractured rocks based on this approach, and the calculation formula and classification criterion of *A_EF_* are expressed as follows [41]:(22)$$A_{EF}=u_{e}^{p}-u_{a}$$
(23)$$A_{EF}<50\;KJ/m^{3}$$
(24)$$50\;KJ/m^{3}\leq A_{EF}\leq 150\;KJ/m^{3}$$
(25)$$150\;KJ/m^{3}\leq A_{EF}\leq 200\;KJ/m^{3}$$
(26)$$A_{EF}>200\;KJ/m^{3}$$

As shown in Figure 18, the rockburst tendency of the rock samples has an upward trend with the increasing loading rate. Specifically, the maximum *A_EF_* of the fractured granite is 500 kJ/m^3^, and the minimum *A_EF_* is 100 kJ/m^3^. The rock *A_EF_* is lower than 200 kJ/m^3^ when the loading rate is lower than 2600 GPa/s, while the occurrence of rockburst is not obvious, and moderate rockburst occurs. The inclination angle of the fracture is also related to the rockburst tendency of the rock sample during the impact process. Only the 45° fractured rock occurs in the low rockburst area, while the 0°, 45°, and 90° fractured rocks almost all have medium and high rockburst, with the rockburst tendency first decreasing and then increasing with the increasing fissure angle, and its magnitude relationship is 0° > 90° > 45°, which also indicates that there is less correlation between rockburst tendency and fracture angle.

#### 5.2.3. Comprehensive Analysis of Rockburst Proneness of Pre-Crack Specimens

Both *M_E_* and *A_E_* reflect the failure degree and residual energy of the fractured rock when the rockburst occurs, respectively. Figure 19 shows the relationship between *M_E_* and *A_EF_* of the rock samples at different fracture angles. *M_E_* gradually increases as *A_EF_* increases. This indicates that *A_EF_* is consistent with the actual rockburst intensity, and it also means that *A_EF_* can accurately reflect the rockburst tendency of rock samples during dynamic loading. Specifically, the rockburst tendency gradually decreases when the fracture angle increases from 0° to 45°. The rockburst tendency increases gradually when the fracture angle increases from 45° to 90°. Therefore, 45° can be regarded as the threshold angle for rockburst proneness. This can be explained because the fracture angle is more than 45° and the effect of the prefabricated fractures reduces the interaction force between the particles and thus generates additional stress, leading to stress concentration inside the rock mass and reducing the intensity of rock-burst. However, releasable elastic energy within the rock during the loading process increases significantly when the fracture angle is greater than 45° and thus enhances the intensity of rockburst.

## 6. Discussion

The sudden release of internal elastic energy during the excavation of deep rock projects may lead to the occurrence of rockburst where the damage process is mainly the release of energy. In essence, rock compression damage is an energy-driven process as the mechanical properties of the rock are changed by the crack fractures, while energy storage is a prerequisite for internal energy. Therefore, studying the law change of the internal elastic energy of rock specimens with loading rates during impact disturbance can reveal the elastic energy generated under the rockburst. However, only analyzing the change law of elastic energy cannot reflect the absolute amount of kinetic energy released from the rock, and further improvements are required. According to the law of conservation of energy, when the rockburst occurs, the kinetic energy of the ejected rock comes from the residual elastic energy after the overall failure of the rock, and the greater the remaining elastic energy, the greater the kinetic energy ejected from the rock fragments and the higher the rockburst tendency is. Therefore, the measure of the intensity of the rock explosion should be judged based on the absolute energy value remaining when the rock is destroyed as an indicator, which provides a theoretical reference for the prevention of rockburst.

The similarities and differences between the kinetic characteristics and energy evolution laws of granites containing prefabricated fractures and general homogeneous rocks are first explored in order to apply the research results to engineering practice. Compared to the intact rocks, there is no significant difference in the final failure mode and energy dissipation law of the fractured rocks, nor are there significant differences in the patterns of elastic and plastic energies that accompany the tests as loading rates increase. The difference lies in their brittleness and resistance to external energy, while the brittleness of fractured rocks is weaker than that of general homogeneous rocks, but the maximum elastic energy stored under the same condition is higher than that of general intact rocks.

The change law of plastic energy, reflected energy, and transmitted energy with loading rate can reveal the degree of damage inside the rock specimen and reflect the degree of development of fissures and cracks in the rock mass. Monitoring the energy of impact stress waves generated during the blasting disturbance and combining this with the variation law of the energy generated during the blasting disturbance can infer the damage degree of the rock mass, which provides a theoretical reference to analyze the integrity of the rock mass and facilitate service blasting mining, rock enclosure support, and stability control of the extraction area. A certain amount of elastic energy is stored inside the rock under the condition of high static stress. Rockburst will occur instantly when the energy exceeds the storage limit of the rock specimen. However, the energy requires impact perturbation to induce a rockburst if the energy is below the energy storage limit of the rock specimen. Although some energy is released during the rockburst, the rock mainly absorbs energy during the whole loading process, which means that the energy absorbed by the rock from the dynamic disturbance is greater than the energy released.

## 7. Conclusions

Many underground engineering projects show that rockburst can occur in flaw rocks under dynamic loading. In this work, dynamic compression tests were conducted on pre-flawed rock specimens using a constrained SHPB system, and the influences of loading rates and flaw angles on deformation properties, fragment distributions, failure patterns, energy storages, dissipations, and surpluses of rocks were analyzed. Moreover, high-speed and crack classification methods were used to study the progressive cracking behavior and final failure form of fractured specimens. The loading rate and fracture angle comprehensively affect the progressive cracking behavior of the rock. The following major conclusions can be drawn:Loading rates can promote the cracking of the rock for a given fracture angle, and only tensile cracks appear when the loading rate is small. Shear cracks become more prominent at higher fracture angles as the loading rate increases, while tensile cracks become more prominent at lower fracture angles. The dominant cracking mechanism changes from tensile to hybrid tensile–shear cracking as the fracture angle increases from 0° to 90°.The three main failure types I-III are tensile failure, X-type shear failure, and tensile–shear mixed failure. The failure type of the specimen is Type I when the crack angle and loading rate are low. Increasing loading rates and crack angles mean the failure modes of other specimens belong to type II and III.The fractal dimension increases with the loading rate, while the average fragment size does the opposite. The fractal dimension decreases as the fracture angle increases, and the average fragment size first increases and then decreases. The increasing loading rate reduces the energy utilization efficiency while it promotes the energy dissipation density for a fixed crack angle. The energy dissipation density first decreases and then increases for a given loading rate, while the energy utilization efficiency first increases and then decreases with the increasing loading ratio. Additionally, all rock specimens feature positive absorbed energy values in the dynamic tests.A a strong ejection phenomenon occurs in the granite specimen under the action of a large loading rate, which means a large number of rock fragments and powders are ejected when it is broken. The rockburst tendency first decreases and then increases with the increasing fracture angle. The *M_E_* values of granite with different fracture angles under impact loading increase with the increasing loading rate, which indicates that the rockburst tendency of rock samples increases with the increasing loading rate.Based on accurate calculations of $u_{d}^{p}$ and $u_{a}^{p}$ for the granite specimens at different crack angles, *A_EF_* is employed to evaluate the rockburst proneness, and the results are in good agreement with the statistical results of *M_E_*. Strong linear relationships exist between *u_e_* and *u_t_*, *u_d_*, and *u_t_* during the shock compression process of granite, and the crack angle does not alter the linear energy storage and dissipation laws of granite.

## Figures and Tables

**Figure 1 materials-15-08920-f001:**
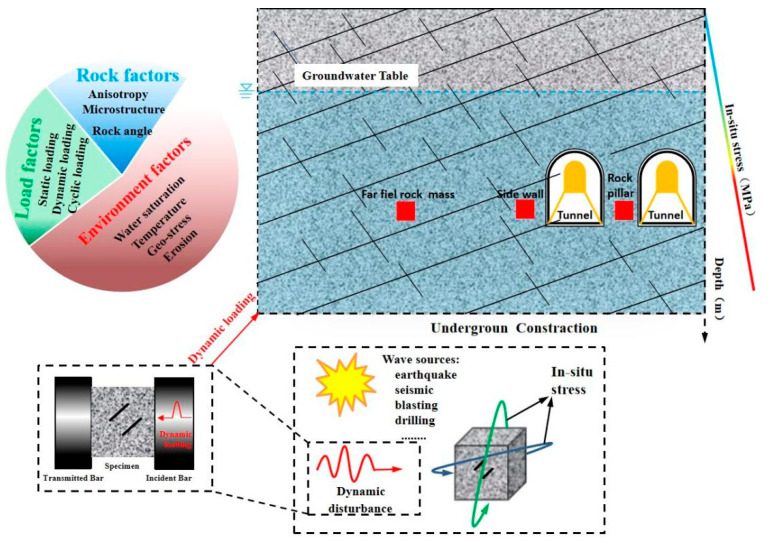
An overview of the influencing factors in rock engineering structures suffering from different impact loads.

**Figure 2 materials-15-08920-f002:**
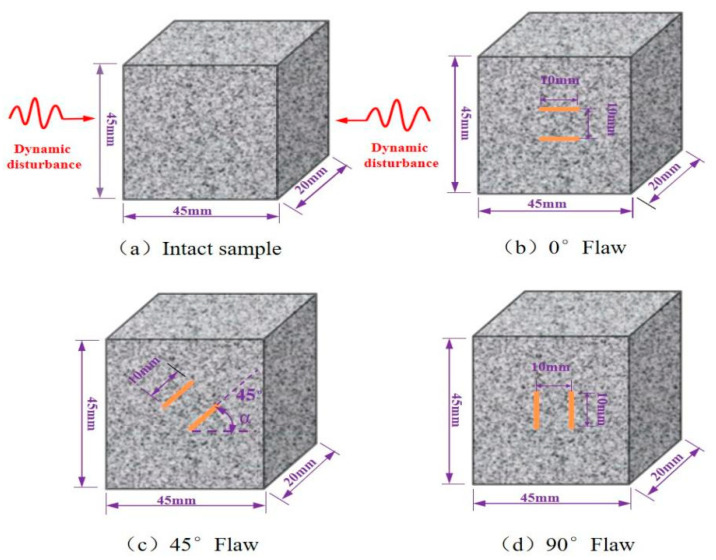
Geometrical images of the specimens.

**Figure 3 materials-15-08920-f003:**
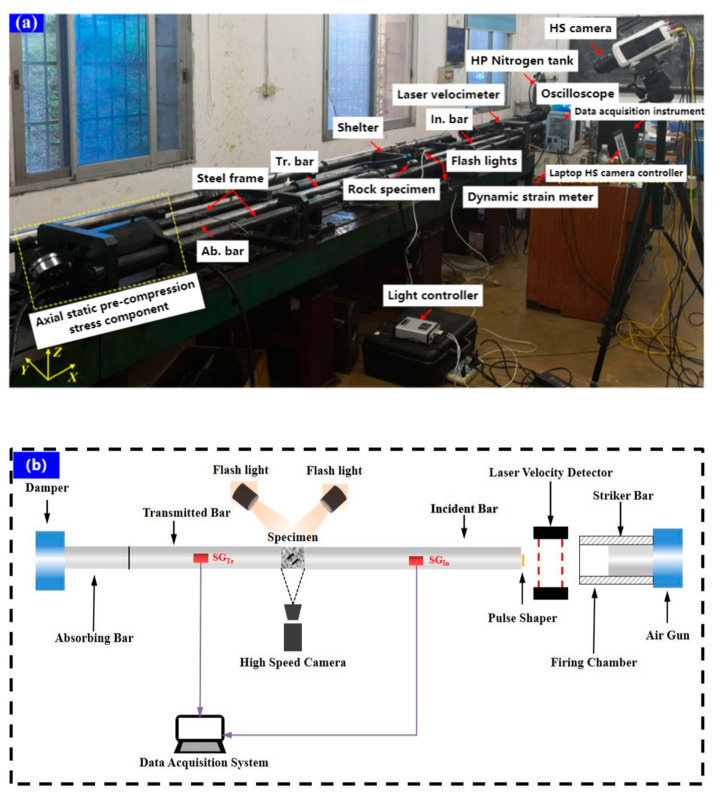
The photographic view of the experimental setup: (**a**) Overview, (**b**) SHPB test system.

**Figure 4 materials-15-08920-f004:**
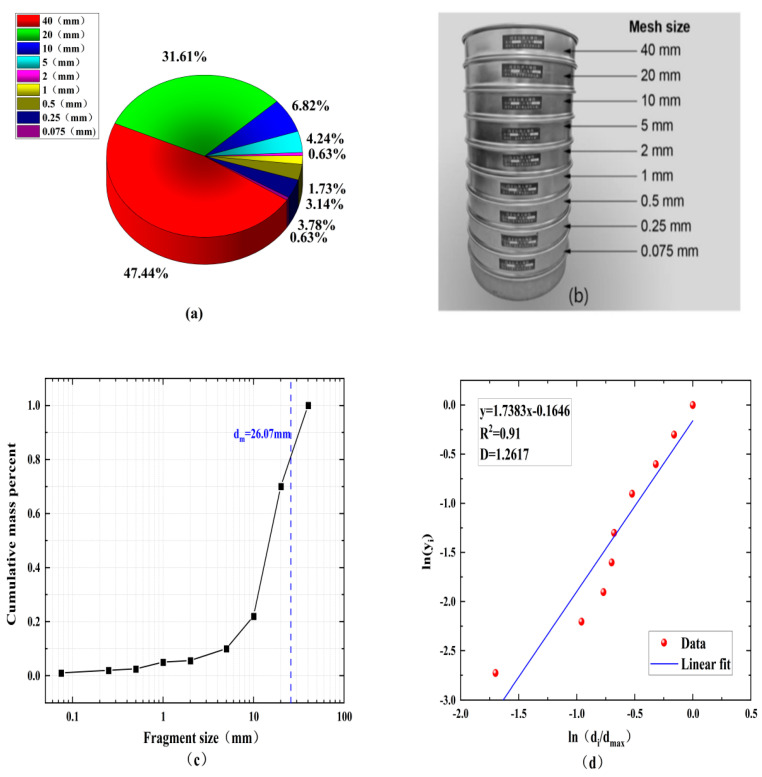
(**a**) A standard sieve used in the sieving tests. (**b**) Sieved fragments with different size intervals. (**c**) A cumulative mass curve against the fragment sizes. (**d**) The fractal dimensions of the rock fragments.

**Figure 5 materials-15-08920-f005:**
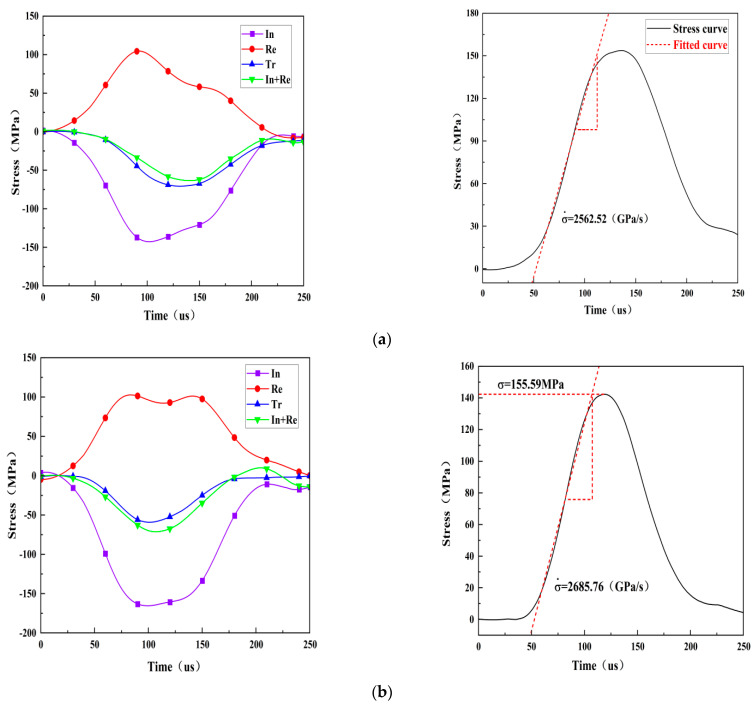
Dynamic stress equilibrium check for the rock specimens with different crack angles: (**a**) rock specimen with a 0° angle, (**b**) rock specimen with a 45° angle, and (**c**) rock specimen with a 90° angle.

**Figure 6 materials-15-08920-f006:**
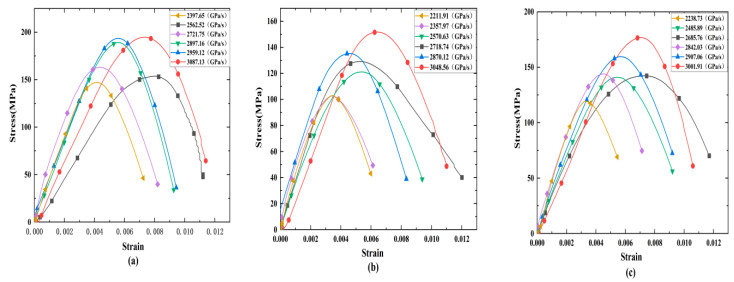
Typical stress–strain relationships of crack specimens: (**a**) rock specimen with a 0° angle, (**b**) rock specimen with a 90° angle, and (**c**) rock specimen with a 90° angle.

**Figure 7 materials-15-08920-f007:**
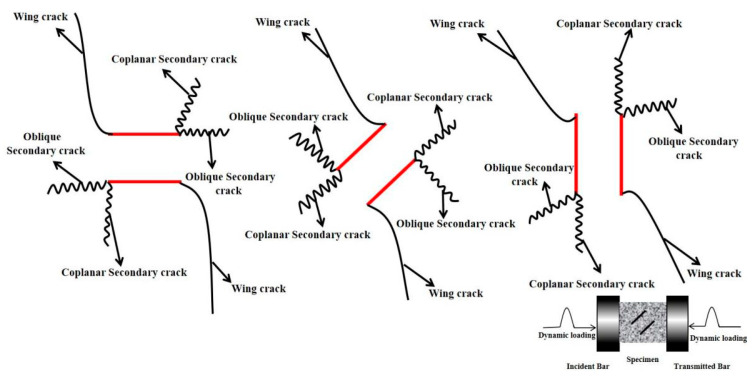
Simplified crack pattern observed in a pre-cracked specimen in dynamic loading.

**Figure 8 materials-15-08920-f008:**
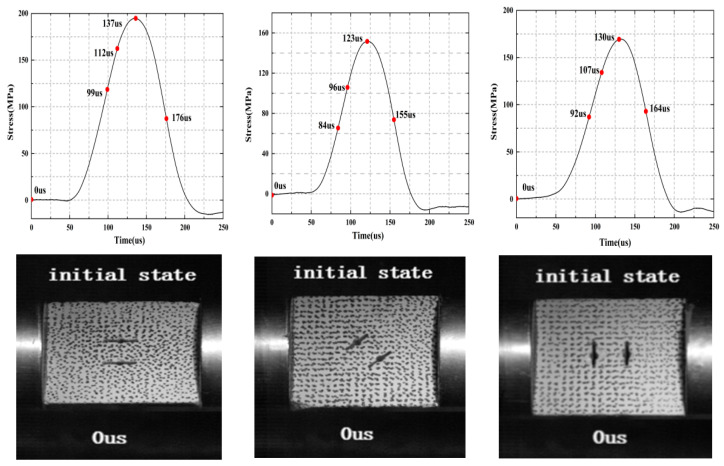
Progressive failure processes of the three typical tested rock specimens with different crack angles: (**a**) rock specimen with a 0° angle, (**b**) rock specimen with a 45° angle, and (**c**) rock specimen with a 90° angle. The right side is the incident rod while the left side is the transmission rod.

**Figure 9 materials-15-08920-f009:**
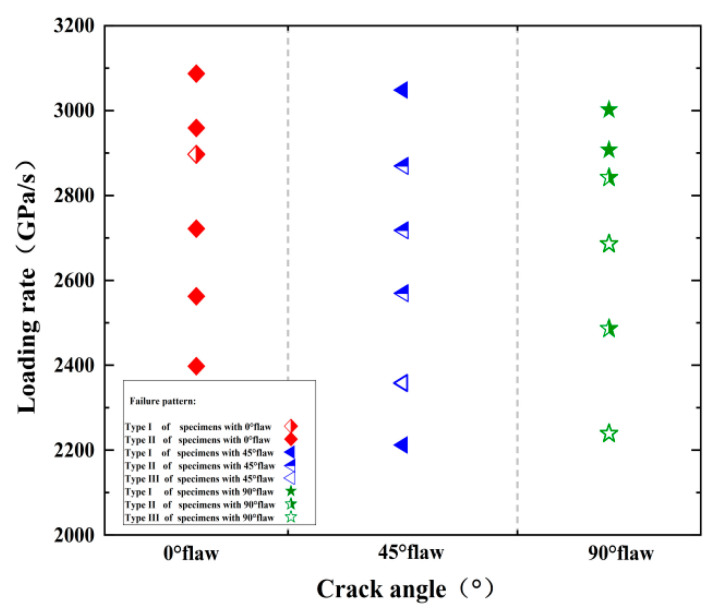
Failure patterns of the flawed specimens under different crack angles and loading rates.

**Figure 10 materials-15-08920-f010:**
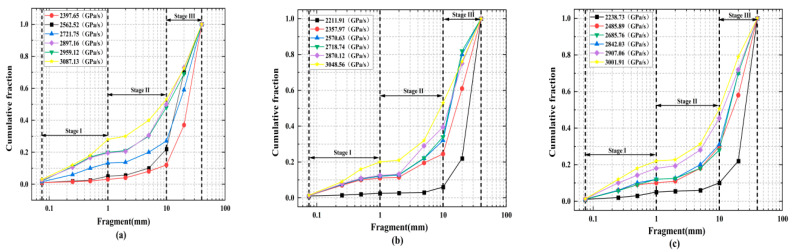
Cumulative mass percent of the pre-cracked rock specimens under different loading rates: (**a**) rock specimen with a 0° angle, (**b**) rock specimen with a 45° angle, (**c**) rock specimen with a 90° angle.

**Figure 11 materials-15-08920-f011:**
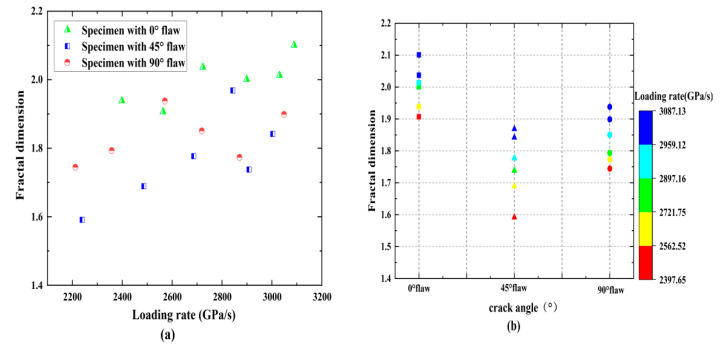
Influences of loading rates and crack angles on the mean fragment size and fractal dimension: (**a**) fractal dimension versus loading rate, (**b**) fractal dimension versus crack angle, (**c**) mean fragment size versus loading rate, and (**d**) mean fragment size versus crack angle.

**Figure 12 materials-15-08920-f012:**
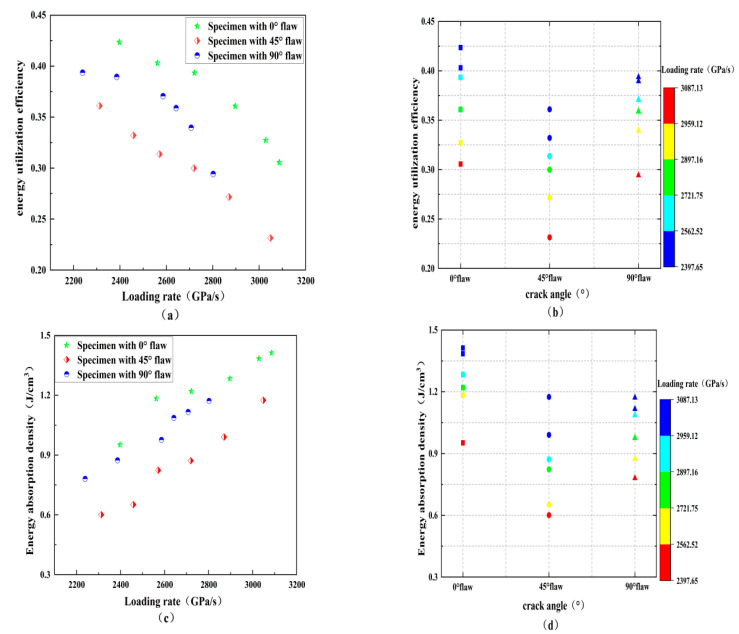
Influences of the loading rate and crack angle on the energy utilization efficiency and energy dissipation density: (**a**) energy utilization efficiency versus loading rate; (**b**) energy utilization efficiency versus crack angle; (**c**) energy dissipation density versus loading rate; and (**d**) energy dissipation density versus crack angle.

**Figure 13 materials-15-08920-f013:**
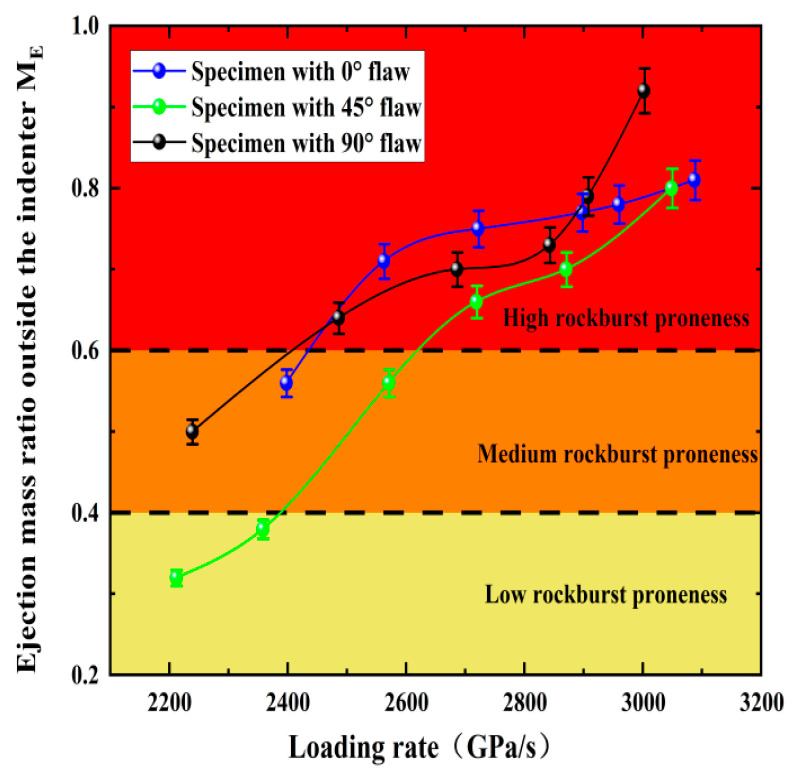
*M_E_* of the pre-crack specimens at different loading rates.

**Figure 14 materials-15-08920-f014:**
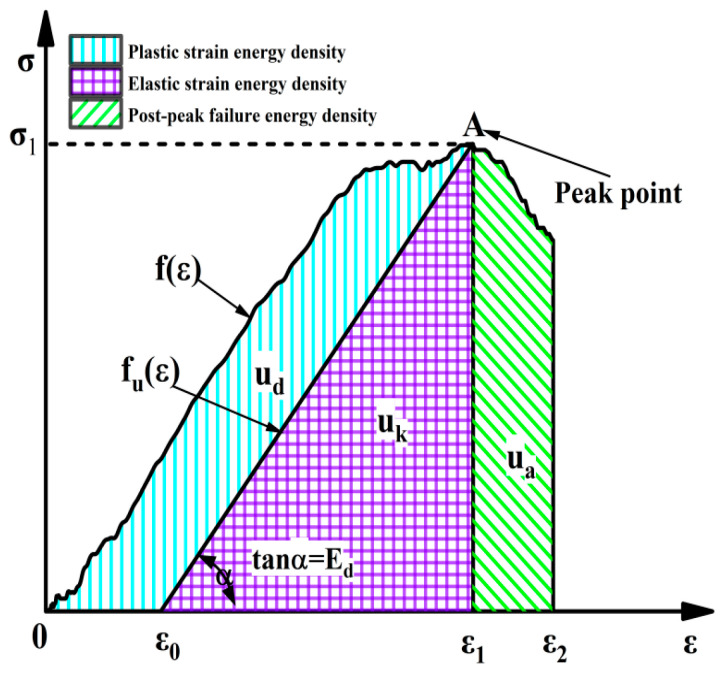
Schematic diagram of calculation for *A_EF_*.

**Figure 15 materials-15-08920-f015:**
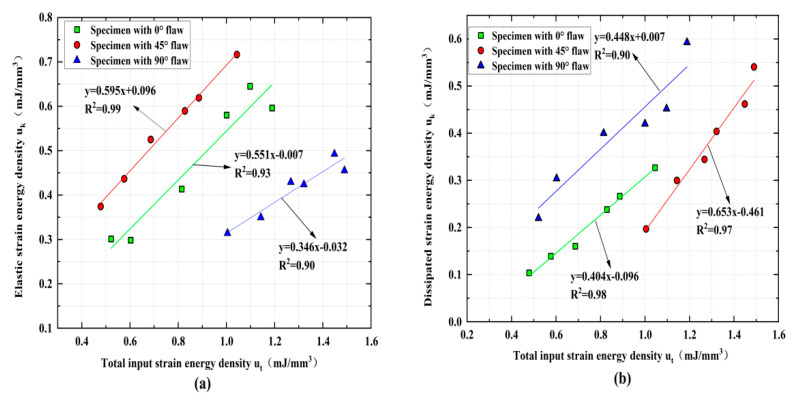
Linear relationships between the three energy densities of pre-crack specimens: (**a**) *u_k_* versus *u_t_* and (**b**) *u_d_* versus *u_t_*.

**Figure 16 materials-15-08920-f016:**
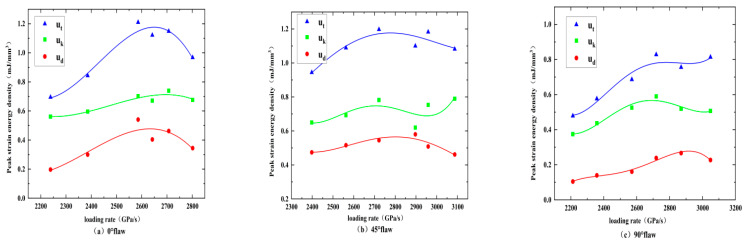
*u_t_*, *u_k_*, and *u_d_* of pre-crack specimens: (**a**) rock specimen with a 0° angle; (**b**) rock specimen with a 45° angle; and (**c**) rock specimen with a 90° angle.

**Figure 17 materials-15-08920-f017:**
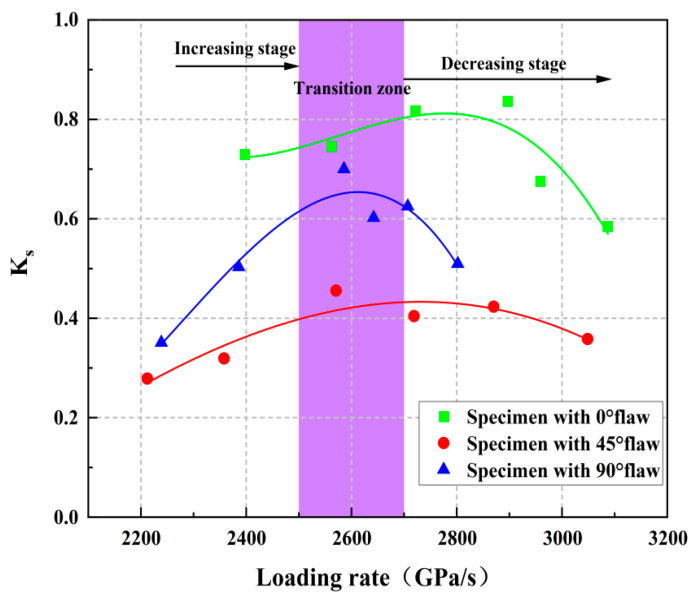
The evolution process of *K_s_* under impact loading.

**Figure 18 materials-15-08920-f018:**
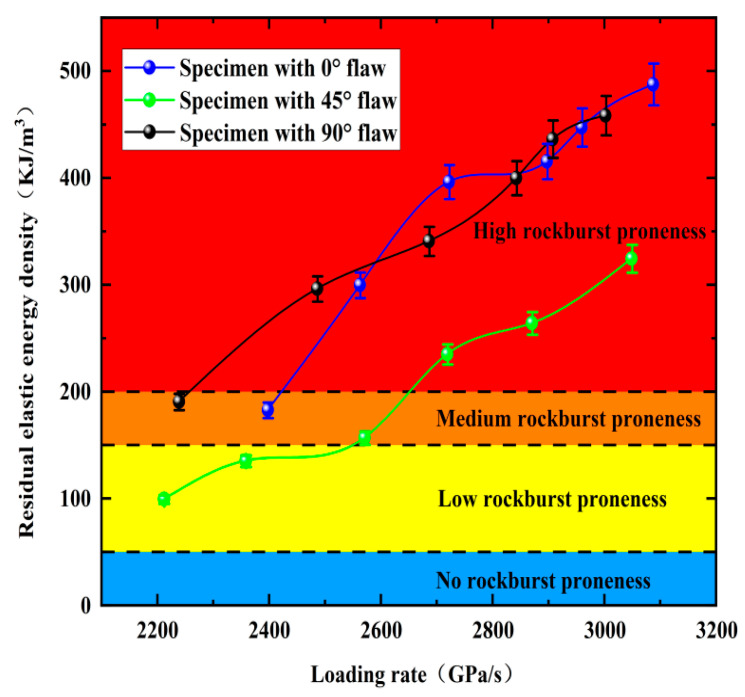
*A_EF_* of pre-crack specimens at different loading rates.

**Figure 19 materials-15-08920-f019:**
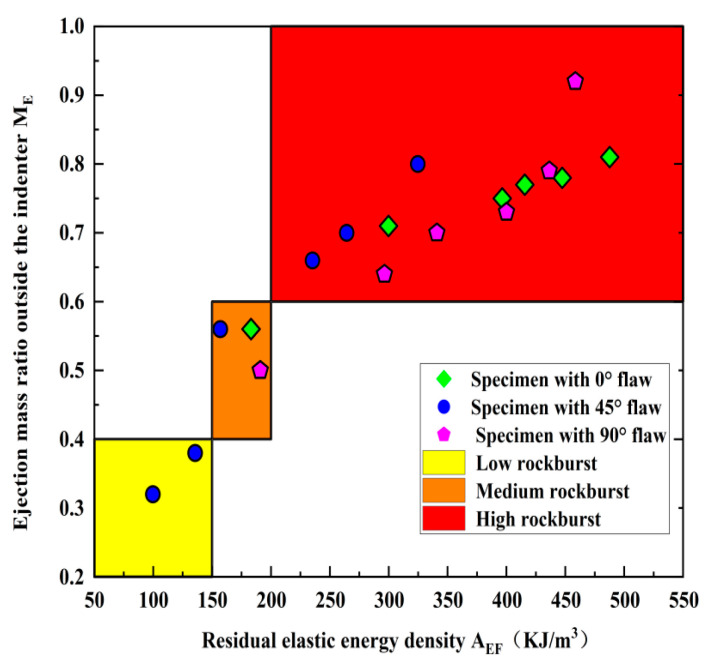
Relationship between the *M_E_* and *A_EF_*.

**Table 1 materials-15-08920-t001:** Comparison of the failure modes of flawed rock specimens with three different inclination angles under distinct loading rates.

	Loading rate (GPa/s)	2397.65	2562.52	2721.75	2897.16	3029.12	3087.13
Angle(°)							
0		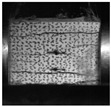	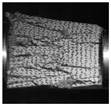	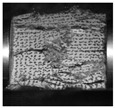	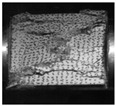	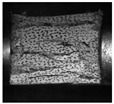	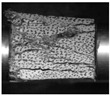
	Loading rate (GPa/s)	2211.91	2357.97	2570.63	2718.74	2870.12	3048.56
Angle(°)							
45		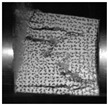	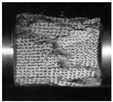	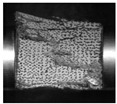	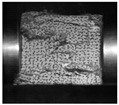	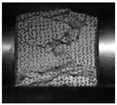	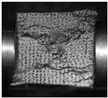
	Loading rate (GPa/s)	2238.73	2485.89	2685.76	2842.03	2907.06	3001.91
Angle(°)							
90		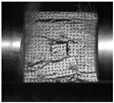	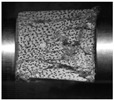	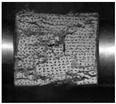	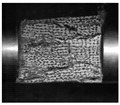	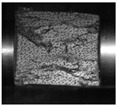	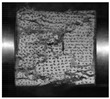

**Table 2 materials-15-08920-t002:** Summary of mean fragment sizes and fractal dimensions of cracked rock specimens under different loading rates.

Specimen	Crack Angles (°C)	Loading Rate (GPa/s)	Mean Fragment Size (mm)	Fractal Dimension
Flaw0°-D1-1	0	2397.65	30.15	1.906
Flaw0°-D2-2	0	2562.52	27.07	1.938
Flaw0°-D3-2	0	2721.75	25.40	2.001
Flaw0°-D4-3	0	2897.16	24.35	2.013
Flaw0°-D5-2	0	2929.12	21.13	2.037
Flaw0°-D6-3	0	3087.13	19.61	2.101
Flaw45°-D1-1	45	2311.91	26.15	1.744
Flaw45°-D2-2	45	2457.97	24.80	1.773
Flaw45°-D3-2	45	2570.63	21.85	1.793
Flaw45°-D4-3	45	2718.74	21.69	1.850
Flaw45°-D5-2	45	2870.12	21.04	1.898
Flaw45°-D6-3	45	3048.56	19.13	1.937
Flaw90°-D1-1	90	2238.73	29.69	1.738
Flaw90°-D2-2	90	2385.89	23.97	1.777
Flaw90°-D3-2	90	2585.76	23.68	1.968
Flaw90°-D4-3	90	2642.03	22.29	1.842
Flaw90°-D5-2	90	2707.06	20.79	1.689
Flaw90°-D6-3	90	2801.91	19.76	1.591

**Table 3 materials-15-08920-t003:** Summary of energy partitions of pre-cracked rock specimens under different loading rates.

Specimen	Crack Angle (°)	Loading Rate (GPa/s)	Energy Utilization	Energy Absorption Density (J/m^3^)
Flaw0°-D1-1	0	2397.65	0.42	0.95
Flaw0°-D2-2	0	2562.52	0.40	1.18
Flaw0°-D3-2	0	2721.75	0.39	1.22
Flaw0°-D4-3	0	2897.16	0.36	1.28
Flaw0°-D5-2	0	2929.12	0.32	1.38
Flaw0°-D6-3	0	3087.13	0.30	1.41
Flaw45°-D1-1	45	2311.91	0.36	0.60
Flaw45°-D2-2	45	2457.97	0.33	0.65
Flaw45°-D3-2	45	2570.63	0.31	0.82
Flaw45°-D4-3	45	2718.74	0.29	0.87
Flaw45°-D5-2	45	2870.12	0.20	0.99
Flaw45°-D6-3	45	2870.12	0.23	1.17
Flaw90°-D1-1	90	2238.73	0.39	0.78
Flaw90°-D2-2	90	2385.89	0.38	0.87
Flaw90°-D3-2	90	2585.76	0.37	0.97
Flaw90°-D4-3	90	2642.03	0.35	1.08
Flaw90°-D5-2	90	2707.06	0.33	1.11
Flaw90°-D6-3	90	2801.91	0.29	1.17

## Data Availability

The data presented in this study are available on request from the corresponding author.

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
