# Peer review of "Experimental Investigation of Pre-Flawed Rocks under Dynamic Loading: Insights from Fracturing Characteristics and Energy Evolution"

_materials, 2022, doi:10.3390/ma15248920_

Round 1

Reviewer 1 Report

The study investigated the effect of dynamic loading rate and fracture orientations on the deformation evolution of granites samples. They propose residual elastic energy index to quantify the rockburst proneness of granite rocks. Overall, the paper is of good quality with lot of laboratory results and conclusions. I would suggest the below point to be addressed.

-          This is not only done in laboratory, you need to cite some field cases to show the possibility to implement such studies in the field. I have proposed some sentences in the attached file, effect of fractures on dissipation energy.

-          It would be good, if the authors add some sentences about the situation if it other type of rocks, other than granite were tested, such as carbonates, what would be the results and conclusions similar or not? And what happened if the samples were saturated, will the results be similar as in this case? Especially in terms of energy dissipation.

-          Give some details about the fractal technique you used to calculate D.

Many comments and corrections are added in the attached document.

Reviewer 2 Report

Main comments:

-The selected loading rates were different for each test. This hinders the comparison. The authors should look for a way to normalize the results in order to improve the comparison.

-How was the transition zone in Figure 16 (onset and end) defined? Does the same applies to the samples with “0” and “90” degrees flaws?

-There is a need for a discussion on the implications of the study, along with a comparison with onsite observations. This should also cover the limitations in terms of testing conditions and specimen size. This is important to translate the lab results to practical engineering situations.

Other comments:

-There is no need to start the 2.1 section stating that all specimens were manufactured following ISRM standards. Instead, state the exact ISRM standard used whenever it is needed in the text.

-Explain the process you followed to create the cracks in the “samples with parallel double cracks”
-The so called tables (like table 1 and 2) are rather figures. Improve the image quality and include them as Figures.

-Reference. Page 3, line 83. “Bobet and LI et al.” must be corrected. This is not the proper way cite two different papers. Both, because these are two papers, and because the “I” in “LI” should be a “small i”.

-Grammar. Page 3, line114. Capital letter in the middle of a sentence. “They found that there are The mechanical …” Same for line 360, “The Table c…”

-Page 4, line 164. “In the process of sample processing…” Revise grammar.

-Fig. 3. Color choice. Change the label style in the photograph as the current yellow color cannot be seen in some sections.

-Page 11, line 331. Which table does the authors refer to? Same as for line 346.

-Page 11, line 333. What is the time unit that the authors refer to? If microseconds, use the proper symbol. Applies to all other sections in the manuscript where the term is used.

-Fig. 8. Include the symbols for the other two angles, 45 and 90 degrees in the figure legend, and remove the background colors. They are distracting.

-Fig. 9. Similarly, remove the background colors. Plot the figures using the same range of cumulative mass fractions, and make sure each subfigure perfectly aligns with the others.

-Fig. 12. Remove the background colors.
